# Tibiofemoral contact force differences between flat flexible and stable supportive walking shoes in people with varus-malaligned medial knee osteoarthritis: A randomized cross-over study

**Scott Starkey**[1], **Rana Hinman**[1], **Kade Paterson**[1], **David Saxby**[2,3], **Gabrielle Knox**[1], **Michelle Hall**[1] *

**1** Centre for Health, Exercise and Sports Medicine, University of Melbourne, Melbourne, Victoria, Australia, **2** Griffith Centre of Biomedical and Rehabilitation Engineering (GCORE), Menzies Health Institute Queensland, Griffith University, Gold Coast, Queensland, Australia, **3** School of Allied Health Sciences, Griffith University, Gold Coast, Queensland, Australia

* halm@unimelb.edu.au

## Abstract

### Objective

To compare the effect of stable supportive to flat flexible walking shoes on medial tibiofemoral contact force (MTCF) in people with medial knee osteoarthritis and varus malalignment.

### Design

This was a randomized cross-over study. Twenty-eight participants aged $\geq$50 years with medial knee osteoarthritis and varus malalignment were recruited from the community. Three-dimensional full-body motion, ground reaction forces and surface electromyograms from twelve lower-limb muscles were acquired during six speed-matched walking trials for flat flexible and stable supportive shoes, tested in random order. An electromyogram-informed neuromusculoskeletal model with subject-specific geometry estimated bodyweight (BW) normalized MTCF. Waveforms were analyzed using statistical parametric mapping with a repeated measures analysis of variance model. Peak MTCF, MTCF impulse and MTCF loading rates (discrete outcomes) were evaluated using a repeated measures multivariate analysis of variance model.

### Results

Statistical parametric mapping showed lower MTCF in stable supportive compared to flat flexible shoes during 5–18% of stance phase (p = 0.001). For the discrete outcomes, peak MTCF and MTCF impulse were not different between the shoe styles. However, mean differences [95%CI] in loading impulse (-0.02 BW·s [-0.02, 0.01], p<0.001), mean loading rate (-1.42 BW·s$^{-1}$ [-2.39, -0.45], p = 0.01) and max loading rate (-3.26 BW·s$^{-1}$ [-5.94, -0.59], p =

**Data Availability Statement:** All relevant data are within the paper and its Supporting Information files.

**Funding:** This project was supported by funding from the Australian National Health and Medical Research Council (NHMRC) (Project Grant #1124418). SCS is supported by an Australian Government Research Training Program Scholarship. MH is supported by an NHMRC Investigator Grant (#1172928). RSH is supported by a NHMRC Senior Research Fellowship (#1154217). KLP is supported by NHMRC Emerging Leadership Investigator Grant (#1174229). The study sponsors did not play any role in the study design, data collection, analysis and interpretation of the data, writing of the manuscript or the decision to submit the manuscript for publication.

**Competing interests:** The authors have declared that no competing interests exist.

0.02) indicated lower measure of loading in stable supportive shoes compared to flexible shoes.

## Conclusions

Stable supportive shoes reduced MTCF during loading stance and reduced loading impulse/rates compared to flat flexible shoes and therefore may be more suitable in people with medial knee osteoarthritis and varus malalignment.

## Trial registration

Australian and New Zealand Clinical Trials Registry (12619000622101).

## Introduction

Knee osteoarthritis (OA), predominantly of the medial tibiofemoral compartment [1], is a highly prevalent and disabling global chronic disease affecting 1 in 5 people over the age of 40 [2]. Pathogenesis of medial knee OA is thought to be due in part to abnormal joint loading within the medial tibiofemoral compartment, with estimates of medial tibiofemoral contact force (MTCF) having been associated with structural disease progression in knee OA [3]. As estimates of tibiofemoral loading are sensitive to frontal plane knee kinematics [4], an important subgroup for consideration is people with medial knee OA and varus malalignment. This subgroup has demonstrated greater functional [5] and structural decline [6] than those with neutrally aligned knees, which is likely due to elevated compressive loads within the medial compartment [7, 8]. As there is no cure for OA, appropriate self-management strategies are strongly advocated to reduce symptoms and delay the need for arthroplasty [9, 10]. Footwear is a low-burden simple self-management strategy.

Current clinical guidelines for knee OA recommend stable supportive shoes be worn [9, 10]. A recent high quality clinical trial by our team showed that stable supportive shoes yielded significant reductions in knee pain compared to flat flexible shoes in people with knee OA [11]. However, biomechanical mechanisms underpinning greater clinical benefits associated with the use of stable supportive shoe compared to flat flexible shoes are unclear. Stable supportive shoes are characterized by high shoe pitch and heel thickness, rigid soles, and motion control properties that limit foot pronation to provide foot stability [12]. Conversely, flat flexible shoes have low shoe pitch and heel thickness, minimal sole rigidity, and are without motion control properties [12].

Currently, the influence of footwear on tibiofemoral loading in knee OA has been inferred from surrogate measures such as the knee adduction moment (KAM) [12–14]. A higher peak KAM has been observed when walking in stable supportive shoes compared to flat flexible shoes in people with knee OA [12–14]. Given higher measures of the KAM have been associated with greater knee OA pain severity [15], the mechanism of knee pain-relieving effects of stable supportive shoes compared to flat flexible shoes is unclear.

The use of the external KAM to infer internal joint load sharing is limited because the KAM does not account for the contribution of muscle loading in its estimates [16–18]. To gain insight into the internal mechanics of the joint, electromyogram (EMG)-informed neuromusculoskeletal modelling can be used, which considers both muscle and external load contributors to estimate MTCF [17]. Although the MTCF during walking is continuous, previous research investigating knee OA biomechanics and footwear has limited their analysis to

discrete measures (i.e., peak KAM or KAM impulse) [12–14, 19], which constitutes substantial data loss and attentional bias. In addition to the magnitude of MTCF, the rate of loading is also of relevance. People with knee OA have been shown to have higher ground reaction force (GRF) loading rates compared to healthy controls [20] and increased KAM loading rate has been correlated with increased medial tibiofemoral cartilage loss [21]. The type of shoe midsole has been shown to influence KAM loading rates [22], and stable and flat flexible shoes differ in sole structure [12]. However, given the limited relationship between the KAM and MTCF [23], the influence of shoe type on MTCF loading rates remains unclear. Thus, further research is needed to investigate the effects of flat flexible and stable supportive shoes on the MTCF across the stance phase of walking and its loading rate, to provide further insight into the bio-mechanical mechanisms underpinning the pain-relieving effects of stable shoes relative to flat flexible shoes.

The aim of this study was to use EMG-informed neuromusculoskeletal modelling to compare the effects of flat flexible and stable supportive walking shoes on MTCF waveforms and discrete measures of MTCF (peak, impulse and loading rate) in people with medial knee OA and varus malalignment. A secondary aim was to compare the effects of flat flexible and stable supportive walking shoes on KAM waveforms and discrete measures of KAM (peak, impulse and loading rate) in the context of existing research.

## Methods

A randomized cross-over study design was used to test the immediate effect of two shoe conditions on parameters of the MTCF during walking. The study was prospectively registered in the Australian and New Zealand Clinical Trials Registry (12619000622101) and is reported according to the items of the TREND checklist applicable to cross-sectional studies [24]. Ethical approval was obtained from the Institutional Human Research Ethics Committee (#1853473) and participants provided their written informed consent prior to testing. The study was conducted at The University of Melbourne.

### Participants

Twenty-eight volunteers were recruited from a clinical trial investigating the 8-week effect of valgus bracing within our laboratory [25]. These participants were recruited from the community in Melbourne, Australia between April 2019 and November 2019 via advertisements in social media and our volunteer database. Data acquisition was performed prior to commencement of this clinical trial. Knee OA was classified according to the American College of Rheumatology clinical and radiographic criteria for knee OA [26]. Participants were included if they: i) were aged 50 years or older; ii) reported knee pain on most days of the past month for >3 months; iii) reported knee pain over the past week while walking of ≥4 on a numerical rating scale; iv) demonstrated radiographic tibiofemoral joint OA (Kellgren & Lawrence grade ≥2) [27]; v) wear a female shoe size 6–11 US or male shoe size 8–13 US (due to the availability of the testing shoes); vi) fit into standard width shoes; and vii) had varus malalignment. Varus malalignment was defined as an anatomic axis angle of <181˚ for females or <183˚ for males [28]. Exclusion criteria were: i) lateral joint space narrowing greater than or equal to medial joint space narrowing; ii) lateral osteophyte grade greater than or equal to medial compartment osteophyte grade; iii) any knee surgery over the past 6 months; iv) awaiting or planning any back or lower-limb surgery over the next 3 months; v) planning to see an orthopaedic surgeon about a knee problem over the next 8 weeks; vi) current or past (3 months) use of oral or intra-articular corticosteroid; vii) systemic arthritis; viii) current or past (6 months) muscular or joint condition other than knee OA; ix) current use of or past (6 months) use of, or

intention to use (next 8 weeks), a knee brace, walking stick or gait aid; and x) unwillingness to wear a knee brace or inability to undergo magnetic resonance imaging (MRI).

## Procedures

A flow-diagram of study procedures is provided in Fig 1. Volunteers were screened via an online survey, followed by telephone screening to confirm eligibility. Potentially eligible participants underwent bilateral knee x-rays if they did not have their own knee x-ray within the previous 12 months. For participants with unilateral symptoms, the unilateral leg was selected as the study knee and data relevant to this limb was analyzed. For participants with bilateral symptoms, the most symptomatic eligible knee was selected as the study knee and data from this limb was analyzed. Participant reported data was collected via REDCap. Biomechanical data was collected by the same researcher within the Centre for Health, Exercise and Sports Medicine gait laboratory at The University of Melbourne.

## Shoe interventions

We selected one style each of commercially available flat flexible and stable supportive shoes that met our existing biomechanical classification criteria [12]. These criteria separate the shoes classes based upon shoe pitch (difference in sole thickness between heel and forefoot), arch support and motion control features, heel height and thickness, sole flexibility, and weight [12]. For the flat flexible condition, we selected Vivobarefoot Primus Lite (men's and women's) and for the stable supportive shoes we selected ASICS Kayano (men's and women's). Shoes were fitted by the same researcher (SCS) for each participant. Once fitted, participants were asked to walk in each shoe for 5 minutes prior to MTCF assessment to ensure familiarization.

## Data acquisition

**Gait analysis.** Participants completed six walking trials for two shoe conditions: stable supportive and flat flexible shoes, performed in a random order. Two of the six trials were used for model calibration purposes (four in total) and the remaining four trials for each condition (eight in total) were for analysis. Participants walked along a 10m walkway at a self-selected speed matched to ±5% across conditions, measured using two photoelectric beams positioned midway along the walkway. Kinematic and ground reaction force data were recorded using a 12-camera motion analysis system (Vicon MX, Oxford Metrics, UK) at 120 Hz and three ground-embedded force plates (AMTI, MASS, USA) at 1200 Hz, respectively. The Vicon system was calibrated using the standardized Vicon 5 reflective marker wand with 10,000 samples across the analysis area. An image threshold error of 0.2 was used for each of the twelve cameras, with recalibration performed if the error exceeded this. A full body marker set, consisting of sixty-seven reflective markers of 14mm diameter were placed on the participant's skin and exterior of shoes according to the University of Western Australia marker set [29]. Surface EMG were acquired at 1200Hz during each walking trial using a wireless telemetered 16-channel Telemyo DTS system (Noraxon, AZ, USA) for twelve lower-limb muscles: tensor fascia latae, gluteus medius, rectus femoris, vastus lateralis, vastus medialis, biceps femoris, semimembranosus, medial gastrocnemius, lateral gastrocnemius, soleus, tibialis anterior and peroneus longus. The skin surface above the muscle belly was prepared and electrodes placed consistent with surface EMG for non-invasive assessment of muscles (SENIAM) guidelines [30]. Participants performed maximum isometric voluntary contraction (MVC) trials to elicit maximal EMG for each of the twelve instrumented muscles in the following positions: (i) seated knee extension, (ii) seated knee flexion, (iii) seated ankle eversion, (iv), seated ankle dorsiflexion, (v) standing hip abduction, and (vi) supported single leg heel raises. Participants

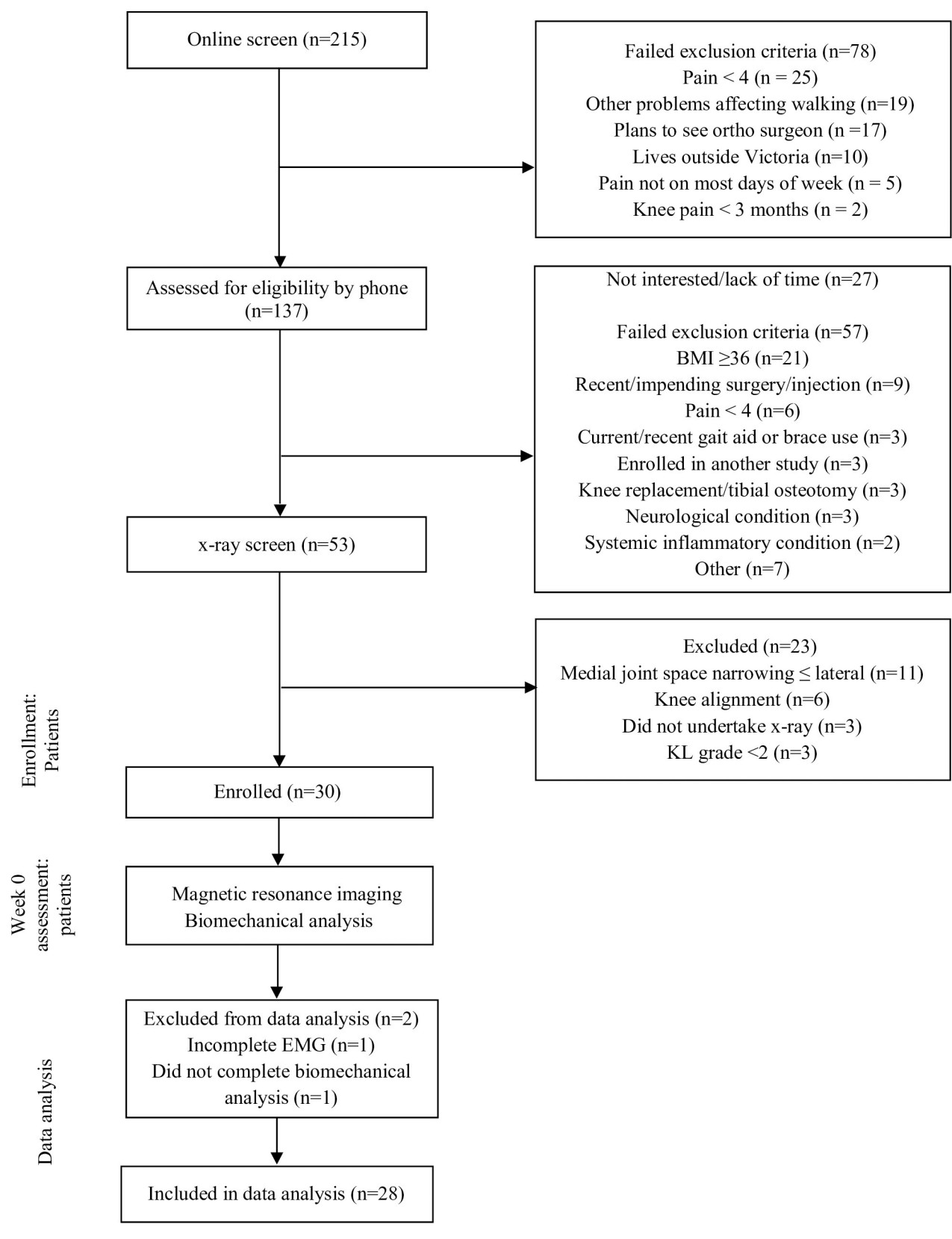

**Fig 1. Flow-diagram of study procedures.**

performed three maximal efforts (5 second duration) for each contraction against resistance with 30 seconds rest in between efforts. For the single leg heel raises, participants held the raised position for three seconds and performed five repetitions. Participants received standardized verbal encouragement to contract maximally during these tasks.

**Magnetic resonance imaging.** A three-dimensional (3D) $T_1$-weighted sagittal volumetric interpolated breath-hold examination (VIBE) of the study knee and a 3D $T_1$-coronal lower-limb scan were undertaken prior to biomechanical assessment using a 3 Tesla MRI machine (Siemens Medical Systems, Erlangen, Germany). The 3D images of the lower limb bones and tibiofemoral joint cartilage were then segmented using Mimics software (Materialise, Leuven, Belgium). These segmentations were used to inform subject-specific anatomical geometry during modelling.

## Data pre-processing

A flowchart outlining the data analysis methods is provided in Fig 2. A force plate threshold of 10 N was used to define heel-strike and toe-off events, obtained using Vicon Nexus software and checked manually by the researcher. Laboratory force plate, marker, and EMG data were pre-processed within Matlab 2019b (MathWorks, Massachusetts, USA) using the MOtoNMS

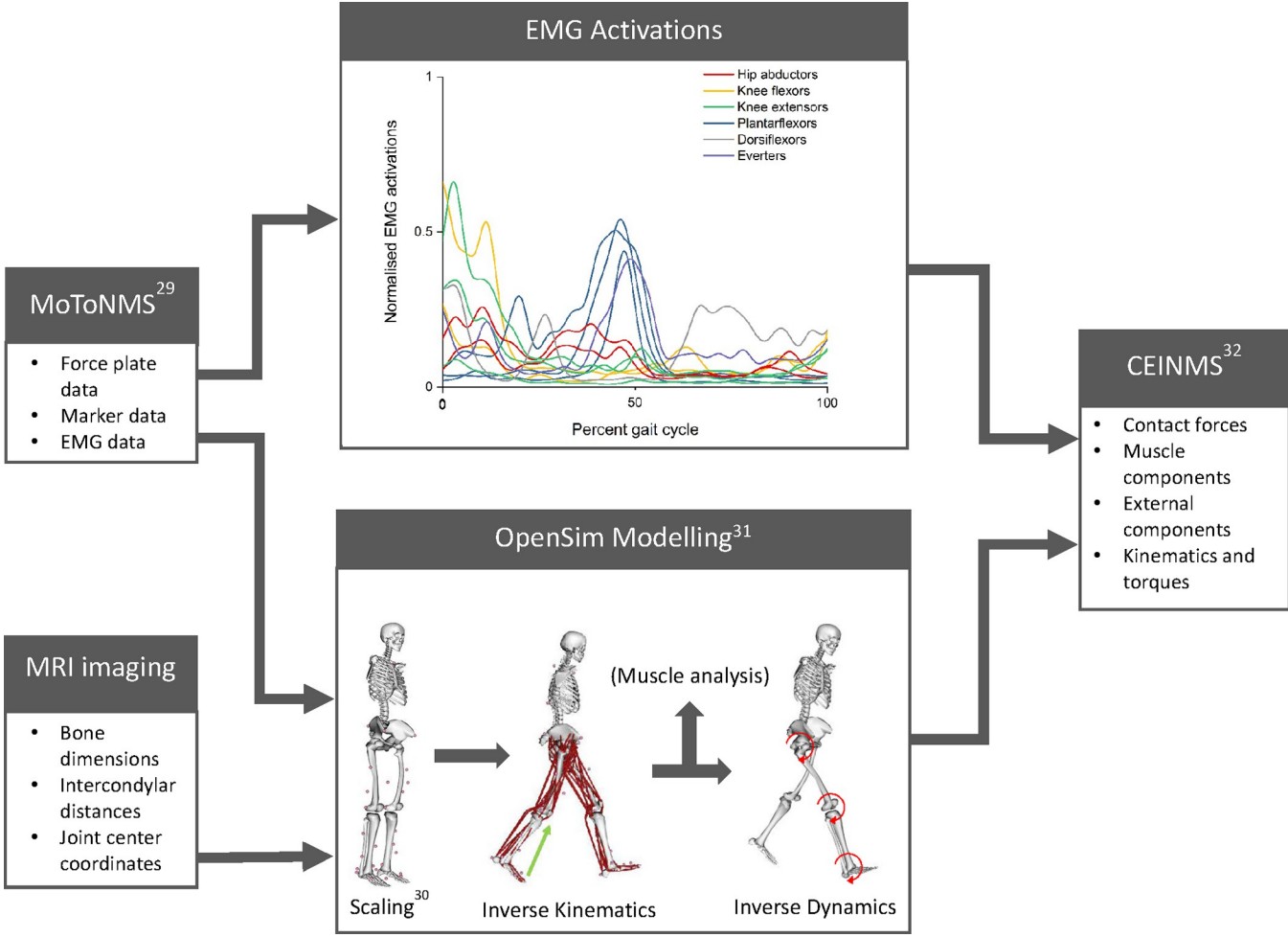

**Fig 2. Flowchart outlining the data processing methods.**

toolbox [31]. This toolbox processes experimental C3D files from Vicon Nexus and coverts the data into OpenSim file formats (i.e., trace and motion files) [32]. The raw EMG data were first band-pass filtered (30–400 Hz), full-wave rectified, then low-pass filtered using zero-lag $2^{nd}$ order Butterworth filter with a 6 Hz low-pass cut-off frequency. The resulting linear envelopes were then amplitude-normalised to the maximum EMG value recorded during the MVC trials.

## Data processing

**Musculoskeletal modelling.**   A generic, full-body musculoskeletal model [33] was used within OpenSim (Version 3.3) [32], which had three rotational degrees of freedom at the hip, one at the knee (with abduction/adduction and internal/external rotations prescribed as a function of knee angle), and one at the ankle. The original model was adjusted to include weightless 'dummy' tibia bodies. This allowed the computation of 3D knee moments and tibio-femoral contact forces, as per existing methods [16]. Contact bodies were added to the medial and lateral compartments of the knee to enable the calculation of net joint moments and mus-cle tendon unit moment arms about each compartment [17]. The locations of these points were determined in 3-matic (Materialise, Leuven, Belgium) by an extrema analysis of the most distal point of the respective femoral condyles. The hip joint centre was obtained in 3-matic as the centre of a sphere fitted on the respective MRI segmented femoral head. Patient-specific scaling of the pelvic, femur and tibia dimensions as well as their mass inertia properties was undertaken by measuring the respective segmented bone lengths and widths using the bony landmarks from the generic model as a reference [33, 34]. Foot and torso model segment dimensions and mass inertia properties were linearly scaled to match individual's anthropom-etry using markers that were acquired during a static pose.

**Calibrated-informed neuromusculoskeletal modelling.**   After model scaling, OpenSim inverse kinematics, inverse dynamics, and muscle analysis tools were used to determine the lower-limb joint kinematics, joint moments, and muscle-tendon unit kinematics, respectively [32]. Model kinematics were low-pass filtered at a 6Hz cut-off frequency [32, 35]. The mod-elled joint moments, muscle-tendon unit kinematics, and processed EMG were then used to calibrate an EMG-informed neuromusculoskeletal model for each participant using the Cali-brated EMG-Informed Neuromusculoskeletal modelling toolbox (CEINMS) [36]. The twelve experimental EMG signals were mapped to twenty muscle-tendon units in the model [36, 37]. For each participant, optimal fibre and tendon slack length of knee-spanning muscles were adjusted using morphometric scaling to preserve their dimensionless operating ranges [38]. Muscle activation dynamics and internal muscle parameters were then functionally calibrated in CEINMS as per our previously reported methods using two of the experimentally acquired walking trials for each condition (four in total) [16]. Following calibration, CEINMS was used in assisted-mode [37] to estimate muscle and tibiofemoral contact forces for the remaining four walking trials for each condition that were not used during calibration. The assisted–mode neural solution synthesised excitation patterns using optimization criteria for knee span-ning muscles that did not have experimental EMG (sartorius and gracilis) and minimally adjusted excitations for muscles with experimental EMG. The knee spanning muscle forces were used as inputs into a planar knee mechanism to estimate MTCF [17]. The absolute mus-cle and external contribution to compartmental tibiofemoral contact force were determined by summing the muscle moments, external torques, and contact reaction moments about the medial and lateral tibiofemoral contact points [16, 17].

**Outcomes.**   The main load variable of interest was MTCF (including muscle and external contribution to MTCF), however we also extracted external knee joint moments (i.e., KAM).

Load variables over each stance phase of gait were spline interpolated to 101 time points and normalised to bodyweight (BW). The peak (BW) and impulse (BW·s) were extracted [18, 39]. The loading phase of gait was defined as the point from initial contact to the first peak of the load variable [21]. The impulse was calculated as time integral of the load variable over the respective time-period. The mean and max loading rate was calculated from existing methods that have evaluated loading rates of the peak vertical GRF [40, 41] and KAM [21, 41]. The mean loading rate ($BW·s^{-1}$) was calculated as the load variable peak divided by the time taken from initial contact to the first load variable peak. Within the same time interval, the max loading rate ($BW·s^{-1}$) was determined by calculating the maximum instantaneous slope of the load variable.

## Sample size calculation

Given that the minimum clinically important difference in MTCF with interventions in OA is unknown, this study utilised a sample of convenience (n = 28) from a bracing trial concurrently undertaken by our research group [25]. A post-hoc sample size calculation determined that this sample was adequately powered to detect a small to medium effect size of 0.35 for change in peak MTCF from barefoot with flat flexible shoes compared to stable supportive shoes. Assuming 80% power, an alpha of 0.05, and a correlation between measurements on the same individual of 0.82 [42], a sample of at least 26 participants was required.

## Statistical analysis

Statistical parametric mapping (SPM) with one-way repeated measures analysis of variance (ANOVA) was conducted within Matlab [43] to examine MTCF waveforms (including muscle and external contributions) and KAM waveforms for the two shoe conditions (flat flexible and stable supportive). SPM is suitable for analysis of MTCF waveforms as it is smooth, sampled above the Nyquist frequency, and bounded by time [43]. If a significant main effect of condition on MTCF or KAM (i.e., suprathreshold cluster existed, $p<0.05$), pairwise comparison with Bonferroni correction was applied to compare the waveforms via SPM {t} maps [44] and reported graphically as mean differences ±95% confidence intervals (95% CI).

Statistical analysis of discrete MTCF and KAM outcomes were conducted using Statistical Package for Social Sciences (SPSS), version 26 (IBM, New York, USA) with significance at $p<0.05$. Dependent variables in the primary analysis were 1) peak MTCF; 2) MTCF impulse; 3) MTCF loading stance impulse; 4) mean MTCF loading rate; 5) max MTCF loading rate; and 6) walking speed. Dependent variables in the secondary analysis were 1) peak KAM; 2) KAM impulse; 3) KAM loading stance impulse; 4) mean KAM loading rate; 5) max KAM loading rate; and 6) walking speed. We compared the differences in the dependent variables using a repeated-measures multivariate analysis of variance (MANOVA), with shoe condition (two levels: stable supportive and flat flexible) as the independent variable. The MANOVA was chosen to control for experiment-wise error rate by evaluating the main effects and interaction of the independent variable on the dependent variables collectively. Assumptions of homogeneity of variance of the residuals, multicollinearity, normal distribution of the residuals and independent observations were evaluated. In the event of a significant main or interaction effect, subsequent post-hoc pairwise comparison with Bonferroni correction was performed to explore significant effects.

## Results

The cohort (n = 28) had slightly more males than females, were overweight, and predominantly had moderate-to-severe radiographic knee OA (Table 1).

**Table 1. Participant characteristics (n = 28).**

| | |
|---|---|
| Age, yr | 63.9 (4.8) |
| Male, n (%) | 16 (57%) |
| Height, m | 1.68 (0.10) |
| Weight, kg | 83.9 (13.6) |
| Body mass index, kg/m$^2$ | 29.6 (3.4) |
| Unilateral symptoms, n (%) | 15 (54%) |
| Duration of symptoms, median (IQR) months | 48 (60) |
| Mean knee pain over the past week, median (IQR)[a] | 6 (2) |
| Most affected leg, right(%) | 22 (79%) |
| Test leg dominant, yes (%) | 25 (89%) |
| Knee alignment[b], degrees | |
| Females | 178.2 (2.6) |
| Males | 178.4 (2.7) |
| Radiographic disease severity grade[c], n (%) | |
| Grade 2 | 9 (32%) |
| Grade 3 | 11 (39%) |
| Grade 4 | 8 (29%) |

Except where indicated otherwise, values are mean (standard deviation)

[a]Numerical rating scale (0 = no pain to 10 = worst pain possible)

[b]Anatomic alignment, where neutral alignment is 181˚ for females and 183˚ for males and varus is <181˚ for females and <183˚ for males; [c]Kellgren-Lawrence grading system; IQR: interquartile range

## Waveform analysis

The results of the primary SPM with one-way repeated measures ANOVA showed significant suprathreshold clusters between the two shoe conditions for MTCF (p = 0.001), muscle contribution to MTCF (p = 0.02) and external contribution to MTCF (p = 0.02) (Fig 3). Post-hoc testing demonstrated that compared to flat flexible shoes, stable supportive shoes had a lower MTCF (at 5–18% of stance), lower muscle contribution to MTCF (at 11–18% of stance), and lower external contribution to MTCF (at 5–15% of stance) (Fig 4).

The results of the secondary SPM with one-way repeated measures ANOVA showed significant suprathreshold clusters between the two shoe conditions for KAM (p<0.001, p = 0.008) (Fig 5). Post-hoc testing demonstrated that compared to flat flexible shoes, stable supportive shoes had a lower KAM at 8–17% of stance and a higher MTCF at 26–51% of stance (Fig 6).

## Discrete measures

The results of the primary repeated measures MANOVA showed a significant main effect between shoe conditions (p<0.001). Subsequent post-hoc analysis found that compared to flat flexible shoes, stable supportive shoes had a lower loading impulse (-0.02 BW·s 95%CI [-0.02, -0.01], p<0.001), lower mean loading rate (-1.42 BW·s$^{-1}$ 95%CI [-2.39, -0.45], p = 0.01) and lower maximal loading rate (-3.26 BW·s$^{-1}$ 95%CI [-5.94, -0.59], p = 0.02) (Table 2). There were no differences observed for peak MTCF (p = 0.48), MTCF impulse (p = 0.12) or walking speed (p = 0.82) (Table 2).

The results of the secondary repeated measures MANOVA showed a significant main effect between shoe conditions (p<0.001). Subsequent post-hoc analysis found that compared to flat flexible shoes, stable supportive shoes had a higher peak KAM (0.04 BW 95%CI [0.02, 0.06], p<0.001) and a higher stance KAM impulse (0.02 BW·s 95%CI [0.01, 0.02], p = 0.001)

 

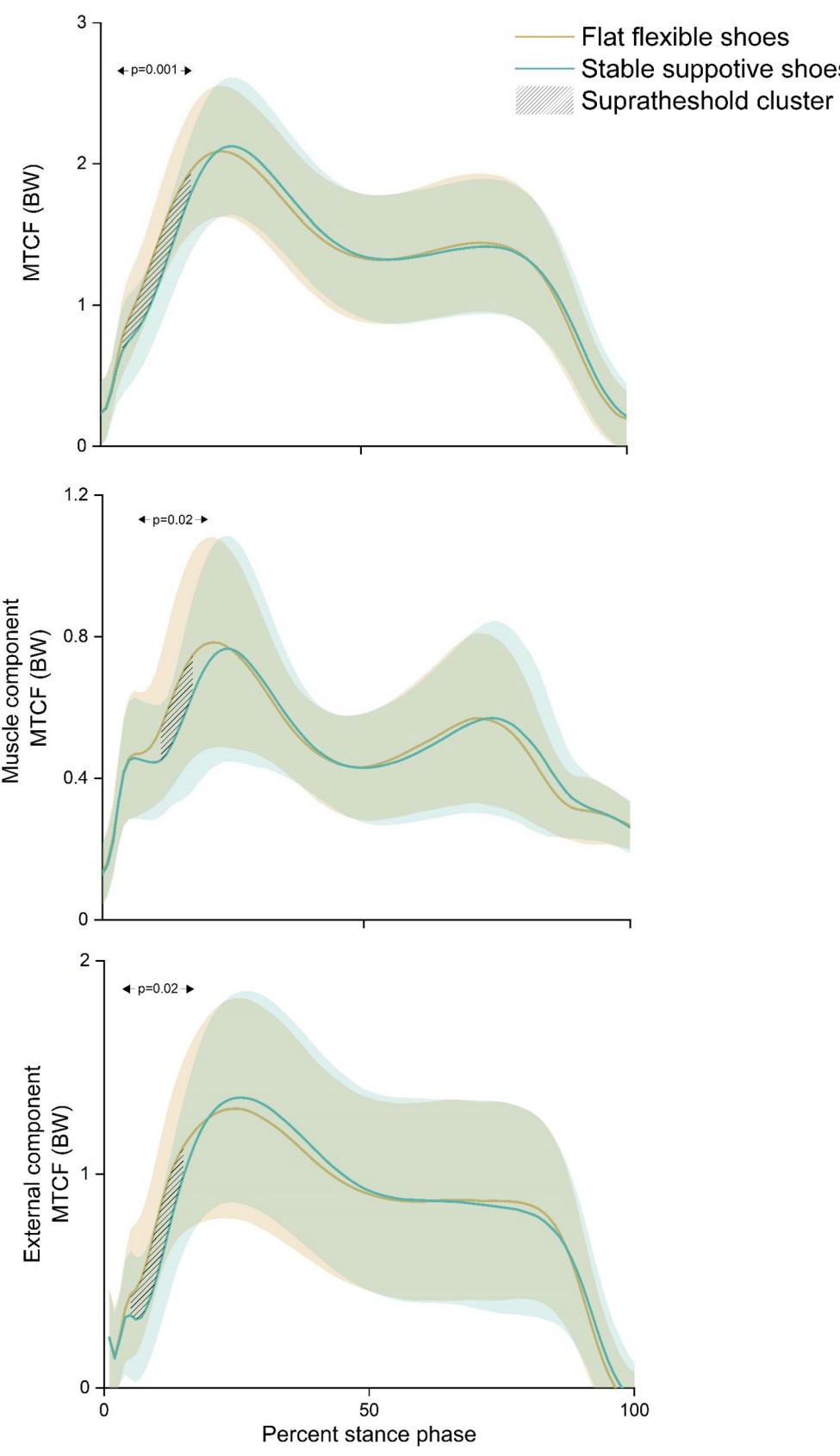

**Fig 3. Ensemble mean (± standard deviation) and statistical parametric mapping (using a one-way repeated measures ANOVA) for medial tibiofemoral joint contact force (top), muscle contribution to medial contact force (middle), and external contribution to medial contact force (bottom) across the stance phase of gait when walking with stable supportive shoes (green) and flat flexible shoes (yellow).**

(Table 3). There were no differences observed for KAM loading stance impulse (p = 0.36), mean KAM loading rate (p = 0.38), max KAM loading rate (p = 0.11) or walking speed (p = 0.68) (Table 3).

## Discussion

The primary objective of this study was to use EMG-informed neuromusculoskeletal modelling to compare the effects of flat flexible to stable supportive walking shoes on MTCF waveforms and discrete measures of MTCF (peak, impulse and loading rate) in people with medial knee OA and varus malalignment. Our SPM of waveform data found the MTCF to be lower in stable supportive shoes compared to flat flexible shoes, but only during the loading period (at 5–18%) of stance. Furthermore, our analysis of discrete variables found a lower MTCF impulse during the loading phase of stance as well as lower loading rates (mean and max) when walking in stable supportive shoes compared to flat flexible shoes, but no differences in peak MTCF or MTCF stance impulse. Collectively, these biomechanical findings indicate that stable supportive shoes may be preferable to flat flexible shoes in people with medial knee osteoarthritis and varus malalignment.

As this is the first study to evaluate the effects of stable supportive and flat flexible shoes on MTCF during walking using EMG-informed neuromusculoskeletal modelling, direct comparison with existing literature is precluded. We observed no statistically significant difference in peak MTCF or MTCF impulse between stable supportive and flat flexible shoes. The mean change from flat flexible shoes for stable supportive shoes in peak MTCF and MTCF impulse was +1.3% and +2.1%, respectively. Given the MDC scores for MTCF using a scaled-generic EMG-driven model is ~12% [42], these differences are unlikely to be meaningful. Secondary analysis of KAM estimates derived from our modelling showed statistically significant increases of 6% in peak KAM and 13% in KAM impulse in stable supportive shoes compared to flat flexible shoes (Table 3). These increases are consistent with existing studies that have used the KAM as a proxy for medial load (e.g., mean peak KAM (+6% (12), +7% (14), +15% (13)) and KAM impulse (+8% (12), +12% (13))). Previous research has suggested that an increase to the KAM does not guarantee increases in MTCF, due to the influence of muscle and ligaments in counteracting external loads [23]. Given the increase in peak KAM in stable supportive shoes compared to flat flexible shoes occurred in absence to changes to peak MTCF, our findings support this concept. Our secondary analysis also found a significant reduction in the KAM with stable supportive shoes compared to flat flexible shoes during the loading period (8%-17%) of stance, but no significant difference in KAM loading rates (mean and max) or KAM loading impulse (Figs 5 and 6, Table 3). We postulate this is due to discrepancies between the peak MTCF and peak KAM across the shoe styles. As ours is the first study to directly evaluate the MTCF loading response using EMG-informed neuromusculoskeletal modelling, further research evaluating this outcome is required to confirm our findings.

Inspection of Fig 3 waveforms suggest that, at peak MTCF, the muscle contribution to MTCF was higher and the external contribution to MTCF lower in flat flexible shoes compared to stable supportive shoes. Based on our findings, we speculate that at peak MTCF there was a concomitant increase in muscle activity across the knee, which nullified the decrease in external loads in flat flexible shoes compared to stable supportive shoes. Limited evidence has

**Stable supportive shoes minus Flat flexible shoes[a]**

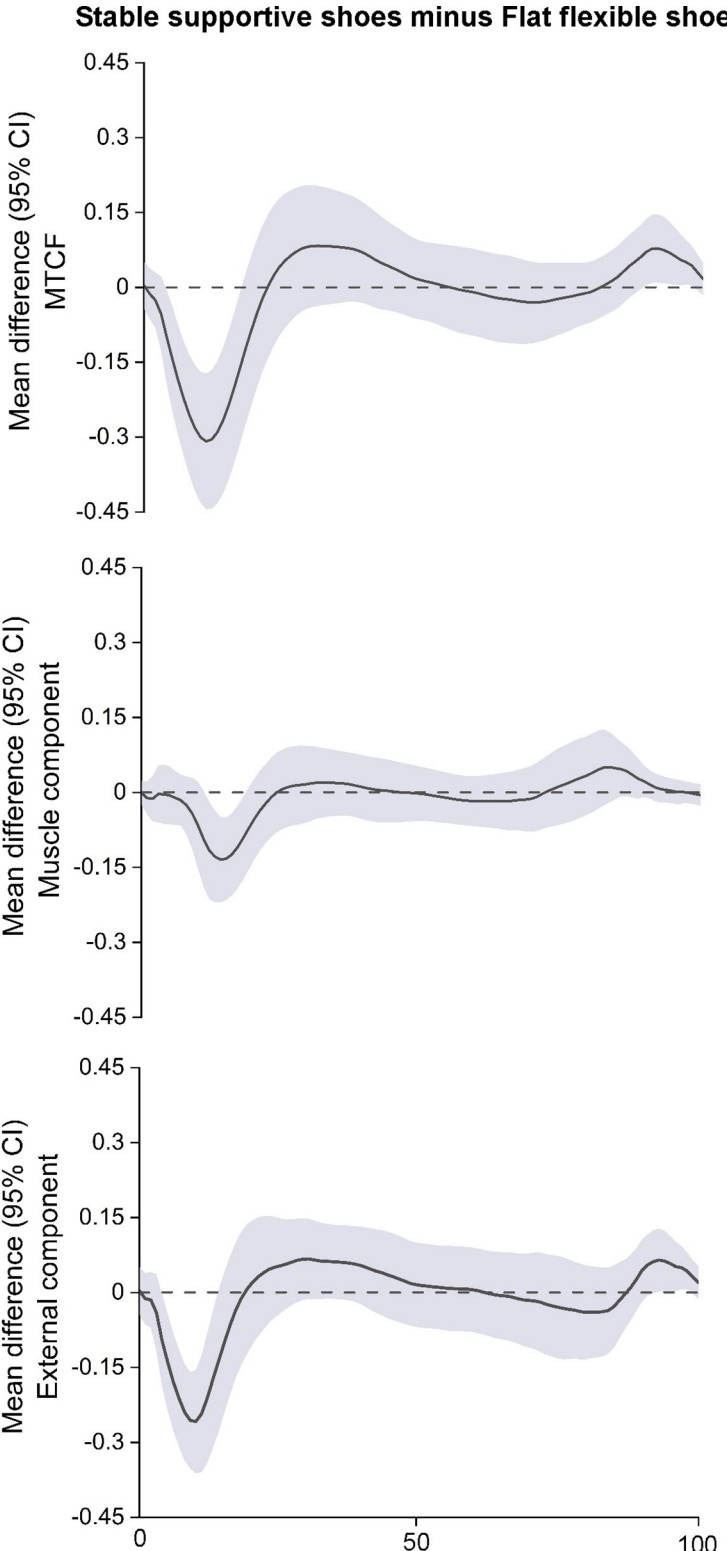

[a] positive values indicate an increase in stable supportive shoes compared to flat flexible shoes;
95%CI = 95% confidence interval; MTCF = medial tibiofemoral joint contact force

**Fig 4. Mean differences (95% CI) across shoe conditions based on ANOVA post-hoc comparison of medial tibiofemoral joint contact force (top), muscle contribution to medial contact force (middle), and external contribution to medial contact force (bottom).**

demonstrated higher medial gastrocnemius and gluteus maximus activation during the stance phase of running in acute bouts of barefoot running compared to shod running [45]. Indeed, it is plausible that flat flexible shoes may increase proximal activation during walking in knee OA. It is also possible that subject-specific biomechanical strategies modulated the MTCF outcomes. This has been demonstrated during real-time modelling of MTCF, where the gait strategies used to alter peak MTCF differed between participants [46]. Furthermore, variations in tibia, rearfoot and forefoot motion have been shown to influence external knee loads [47]. However, the influence of flat flexible and stable supportive shoes on these parameters were not evaluated in this study. Further research may be necessary to investigate potential subgroup or individual biomechanical and muscle activation strategies in people with knee OA.

During the loading phase of stance, there is a large, high frequency impact force through the lower limb. The shock absorption capacity of this impact force (loading rate) has important clinical implicaitons [40, 48]. Compared to healthy controls, higher rates of GRF loading have

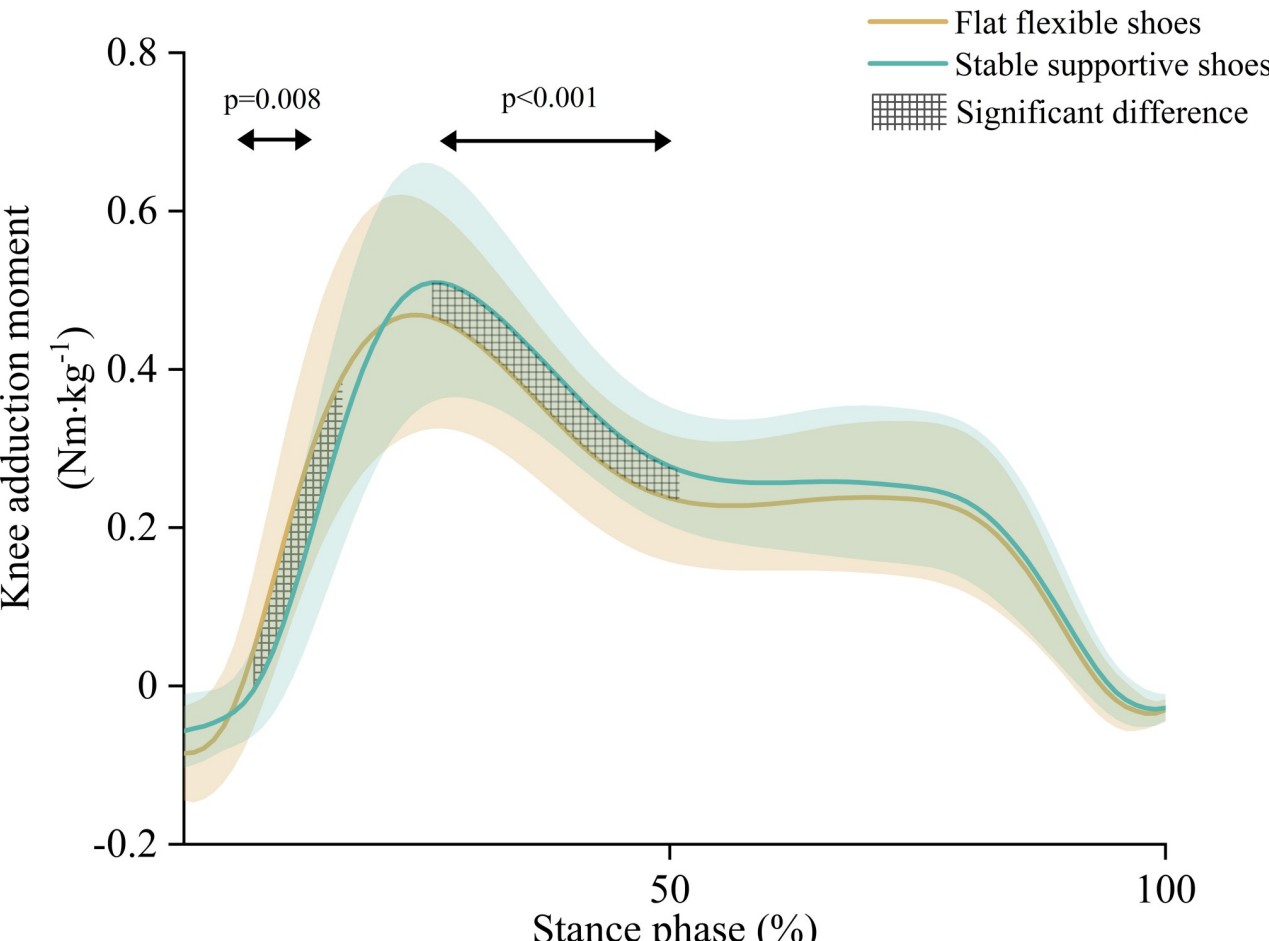

**Fig 5. Ensemble mean (± standard deviation) and statistical parametric mapping (using a one-way repeated measures ANOVA) for external knee adduction moment (BW) across the stance phase of gait when walking with stable supportive shoes (green) and flat flexible shoes (yellow).**

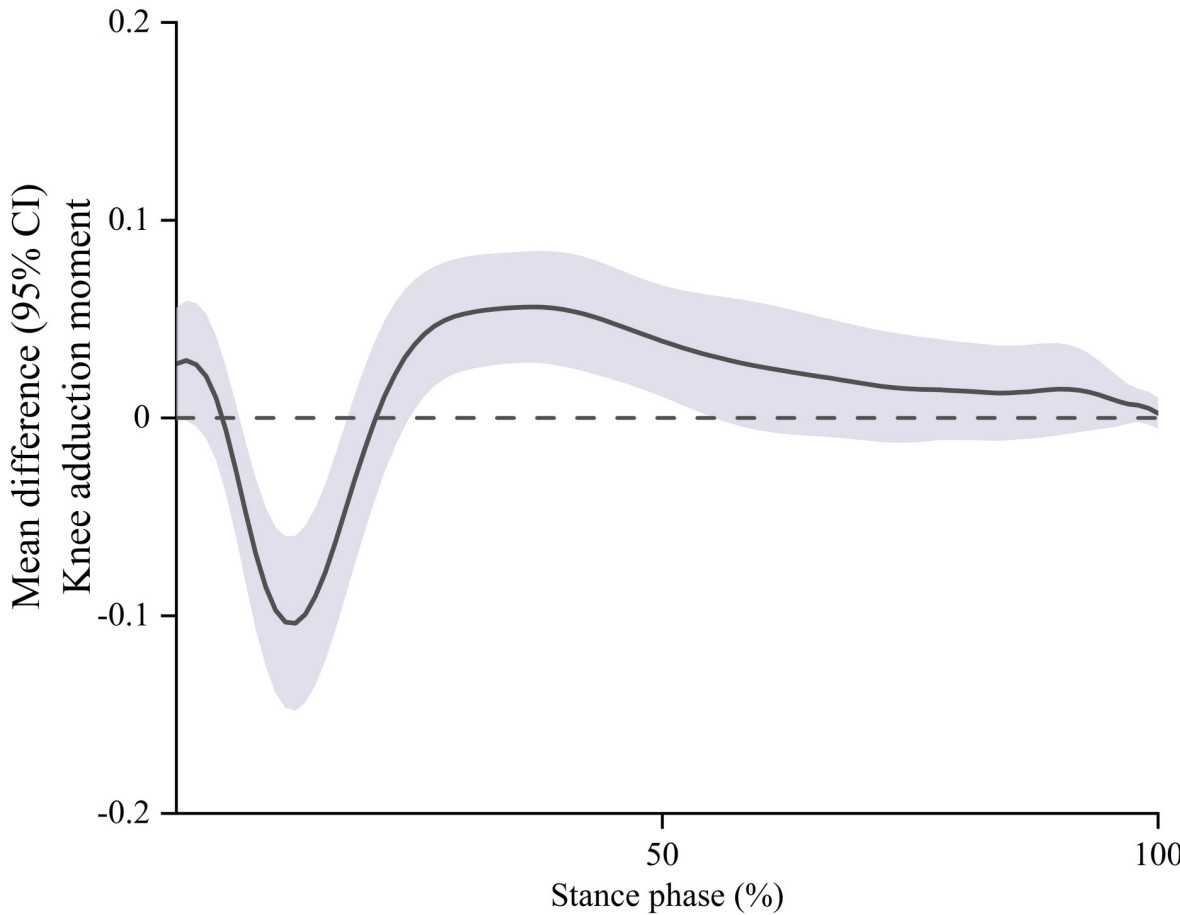

## Stable supportive shoes minus Flat flexible shoes[a]

[a] positive values indicate an increase in stable supportive shoes compared to flat flexible shoes;
95%CI = 95% confidence interval;

**Fig 6. Mean differences (95% CI) across shoe conditions based on ANOVA post-hoc comparison of external knee adduction moment (BW).**

been associated with the presence of knee OA [20] and increased loading rate of the external KAM has been correlated with increased medial tibiofemoral cartilage loss on MRI [21]. Walking in stable supportive shoes resulted in a lower MTCF loading rate compared to flat flexible shoes, which appears to be driven by a rightward shift in the stable supportive MTCF waveform shortly after heel-strike (Fig 3). These findings are intuitive considering the differences in shoe design. Stable supportive shoes are designed with thick, high pitched heels and cushioned soles, whereas flat flexible shoes have thin, low pitched heels and minimal shoe cushioning. Unfortunately, other research comparing the effects of stable supportive and flat flexible shoe styles on loading rates is limited to running in young, healthy populations [22], which makes direct comparisons with our findings in walking difficult given that loading rates during running are influenced by both foot strike pattern [49–51] and footwear style [20, 50]. Furthermore, it is challenging to compare a clinical population aged ≥50 years with symptomatic medial knee OA and varus malalignment to a young population with asymptomatic knees due to disease-specific biomechanical strategies [20, 25].

**Table 2. Mean change (SD) in spatiotemporal and discrete joint contact force variables for flat flexible and stable supportive shoes, with accompanying mean differences (95% confidence intervals (CI)).**

| | Flat flexible (n = 28) | Stable supportive (n = 28) | Mean difference (95%CI) Stable supportive minus Flat flexible[a] |
|---|---|---|---|
| *Spatiotemporal* | | | |
| Walking speed (m·s$^{-1}$) | 1.29 (0.15) | 1.29 (0.15) | 0.00 (-0.01, 0.02) |
| *Joint contact forces (BW)* | | | |
| Peak medial | 2.13 (0.46) | 2.16 (0.49) | 0.03 (-0.05, 0.11) |
| *Joint contact impulse (BW·s)* | | | |
| Loading stance | 0.10 (0.02) | 0.08 (0.02) | -0.02 (-0.02, -0.01) |
| Overall stance | 0.89 (0.24) | 0.91 (0.25) | 0.02 (-0.01, 0.04) |
| *Loading rate (BW·s$^{-1}$)* | | | |
| Mean | 14.03 (3.84) | 12.61 (3.28) | -1.42 (-2.39, -0.45) |
| Max | 29.01 (12.63) | 25.74 (12.19) | -3.26 (-5.94, -0.59) |

[a] positive values indicate an increase in stable supportive shoes compared to flat flexible shoes

A novel insight from our use of neuromusculoskeletal modelling is the evaluation of the absolute muscular and external load contributions to MTCF. During loading stance, there was a decrease in both the muscle and external load contributions to MTCF during walking with stable supportive shoes compared to flat flexible shoes (Fig 4). The difference in the muscular contribution to MTCF was observed between 11% to 18% of stance, while the difference in external contribution to MTCF was evident between 5% to 15% of stance (Fig 4). At heel-strike, shoes increase both ankle dorsiflexion and knee flexion compared to barefoot, which likely increases gastrocnemius and quadriceps activation [52]. It is plausible that the comparable waveforms of muscle contributions to MTCF between 1%-10% of stance across our footwear conditions are a result of similar muscle activation profiles stemming from wearing shoes of any type. However, without a barefoot control comparison these inferences lack validation.

This study focused on comparing immediate effects of two shoe styles. Interestingly, "athletic", "sturdy/supportive" and "cushioned" shoes are most frequently worn by people with knee OA, with both clinicians and patients believing that they are the most suitable [53]. However, these characteristics are not mutually exclusive to stable supportive shoes. A rigorous RCT comparing stable supportive to flat flexible shoes reported 88% of participants wore shoes with mixed properties before enrolment and had comparable treatment expectations at

**Table 3. Mean change (SD) in spatiotemporal and discrete knee adduction moment variables for flat flexible and stable supportive shoes, with accompanying mean differences (95% confidence intervals (CI)).**

| | Flat flexible (n = 28) | Stable supportive (n = 28) | Mean difference (95%CI) Stable supportive minus Flat flexible[a] |
|---|---|---|---|
| *Spatiotemporal* | | | |
| Walking speed (m·s$^{-1}$) | 1.29 (0.15) | 1.29 (0.15) | 0.00 (-0.01, 0.02) |
| *Knee adduction moment (BW)* | | | |
| Peak | 0.49 (0.15) | 0.52 (0.15) | 0.04 (0.02, 0.06) |
| *Knee adduction moment impulse (BW·s)* | | | |
| Loading stance | 0.04 (0.01) | 0.04 (0.01) | 0.00 (0.00, 0.00) |
| Overall stance | 0.15 (0.05) | 0.17 (0.05) | 0.02 (0.01, 0.02) |
| *Knee adduction moment loading rate (BW·s$^{-1}$)* | | | |
| Mean | 3.10 (1.29) | 2.98 (1.04) | -0.11 (-0.36, 0.15) |
| Max | 7.29 (2.44) | 6.71 (1.78) | -0.55 (-1.25, 0.14) |

[a] positive values indicate an increase in stable supportive shoes compared to flat flexible shoes

baseline [11]. Based on these findings, it is unclear whether the lower MTCF with stable supportive shoes observed in this study stems from habitual walking in this shoe type. Indeed, study of longitudinal responses of MTCF to footwear needs to be undertaken to understand any muscular, gait or habitual adaptations.

Previous research has demonstrated relationships between KAM (peak and impulse) and knee OA pain [15]. Despite being associated with a higher KAM relative to flat flexible shoes [12–14], robust clinical trial evidence shows that stable supportive shoes provide greater knee pain relief compared to flat flexible shoes [11]. This suggests that factors other than the KAM contribute to changes in knee pain, and/or reflect the limitations of the KAM as a proxy of medial tibiofemoral joint loads during walking. We applied an EMG-informed model to measure medial tibiofemoral loads, as these methods provide more valid indicators of in-vivo MTCF than the KAM [54]. We found no difference in peak MTCF, or MTCF impulse throughout stance, between flat flexible shoes and stable supportive shoes but observed reduced measures of loading with supportive shoes compared to the flexible shoes. Taken together, our exploratory hypothesis-generating findings suggest that greater knee OA pain-relief associated with stable supportive shoes compared to flat flexible shoes may be due to reduced MTCF loading rates.

This is the first study to apply a validated EMG-informed neuromusculoskeletal model to compare the effects of shoe conditions on MTCF, in any population. We implemented subject-specific lower-limb joint geometry, which has previously been shown to improve predictions of MTCF [34]. Furthermore, the root mean square error values between CEINMS generated knee flexion/extension torques and those from inverse dynamics were well within best practice and CEINMS guidelines [36, 55] and did not differ significantly between conditions (S1 Table). This demonstrates the modelling has well satisfied the primary rigid body physics while also respecting the unique subject- and task-specific muscle activation patterns. However, some limitations do warrant consideration. First, these findings are only generalisable to participants with knee OA who also have varus knee malalignment. Second, our study sample included slightly more males than females, however knee OA is more prevalent in women than men [56]. Lastly, extensive validation of our EMG-informed neuromusculoskeletal modelling is hindered by limited datasets to directly validate MTCF predictions [34].

Our findings suggest that walking in stable supportive shoes reduced MTCF during loading phase of stance compared to flat flexible shoes in people with medial knee OA and varus malalignment. Stable supportive shoes also reduced mean and maximal loading rate, and reduced loading impulse compared to flat flexible shoes, but not peak MTCF or MTCF stance impulse. Stable supportive shoes may therefore be preferred over flat flexible shoes in people with medial knee OA and varus malalignment.

## Supporting information

**S1 Checklist. TREND statement checklist.**
(PDF)

**S1 Table. Root mean square error (Nm·kg) for CEINMS predicted knee flexion/extension moments compared to OpenSim generated inverse dynamics.** Data reported as (Mean (SD) of groups and mean (95% CI) change within and between groups, adjusted for baseline scores.
(DOCX)

**S1 File. Stance phase medial tibiofemoral contact force for flat flexible and stable supportive shoes.**
(CSV)

**S2 File. Stance phase muscle contribution to medial tibiofemoral contact force for flat flexible and stable supportive shoes.**
(CSV)

**S3 File. Stance phase external contribution to medial tibiofemoral contact force for flat flexible and stable supportive shoes.**
(CSV)

**S4 File. Walking speed, peak medial tibiofemoral joint contact force, loading and stance phase medial joint contact impulse and loading rate (mean and max) for flat flexible and stable supportive shoes.**
(CSV)

**S5 File.**
(PDF)

## Acknowledgments

CEINMS is being developed by the Rehabilitation Engineering Group at University of Padua, Italy, and the Menzies Health Institute Queensland, Griffith University, Queensland, Australia.

## Author Contributions

**Conceptualization:** Rana Hinman.

**Data curation:** Scott Starkey, Gabrielle Knox, Michelle Hall.

**Formal analysis:** Scott Starkey, Rana Hinman, Kade Paterson, David Saxby, Michelle Hall.

**Funding acquisition:** Rana Hinman, Michelle Hall.

**Investigation:** Scott Starkey, Gabrielle Knox, Michelle Hall.

**Methodology:** Scott Starkey, Rana Hinman, Kade Paterson, David Saxby, Michelle Hall.

**Project administration:** Scott Starkey, Gabrielle Knox.

**Resources:** Rana Hinman, Michelle Hall.

**Software:** David Saxby.

**Supervision:** Rana Hinman, Kade Paterson, David Saxby, Michelle Hall.

**Validation:** Scott Starkey, David Saxby.

**Writing – original draft:** Scott Starkey.

**Writing – review & editing:** Rana Hinman, Kade Paterson, David Saxby, Gabrielle Knox, Michelle Hall.

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
