## [Decision Letter · Decision Letter 0]

4 Jan 2022

PONE-D-21-29346

The immediate effect of flat flexible vs stable supportive walking shoes on tibiofemoral contact force in people with varus-malaligned medial knee osteoarthritis

PLOS ONE

Dear Dr. Hall,

Thank you for submitting your manuscript to PLOS ONE. After careful consideration, we feel that it has merit but does not fully meet PLOS ONE’s publication criteria as it currently stands. Therefore, we invite you to submit a revised version of the manuscript that addresses the points raised during the review process.

While the reviewer 1 has not find major concerns, the reviewer 2 pointed out some concerns that deserve attention.

We look forward to receiving your revised manuscript.

Kind regards,

Leonardo A. Peyré-Tartaruga, Ph.D.

Academic Editor

PLOS ONE

https://journals.plos.org/plosone/s/file?id=ba62/PLOSOne_formatting_sample_title_authors_affiliations.pdf”

“This project was supported by funding from the Australian National Health and Medical Research Council (NHMRC) (Project Grant #1124418). SCS is supported by an Australian Government Research Training Program Scholarship. MH is supported by an NHMRC Investigator Grant (#1172928). RSH is supported by a NHMRC Senior Research Fellowship (#1154217). KLP is supported by NHMRC Emerging Leadership Investigator Grant (#1174229). The study sponsors did not play any role in the study design, data collection, analysis and interpretation of the data, writing of the manuscript or the decision to submit the manuscript for publication.”

“This project was supported by funding from the Australian National Health and Medical Research Council (NHMRC) (Project Grant #1124418). SCS is supported by an Australian Government Research Training Program Scholarship. MH is supported by an NHMRC Investigator Grant (#1172928). RSH is supported by a NHMRC Senior Research Fellowship (#1154217). KLP is supported by NHMRC Emerging Leadership Investigator Grant (#1174229). The study sponsors did not play any role in the study design, data collection, analysis and interpretation of the data, writing of the manuscript or the decision to submit the manuscript for publication.”

Reviewers' comments:

Reviewer's Responses to Questions

**Comments to the Author**

1. Is the manuscript technically sound, and do the data support the conclusions?

Reviewer #1: Yes

Reviewer #2: Yes

2. Has the statistical analysis been performed appropriately and rigorously? 

Reviewer #1: Yes

Reviewer #2: Yes

3. Have the authors made all data underlying the findings in their manuscript fully available?

Reviewer #1: No

Reviewer #2: No

4. Is the manuscript presented in an intelligible fashion and written in standard English?

Reviewer #1: Yes

Reviewer #2: Yes

5. Review Comments to the Author

Reviewer #1: A randomized clinical trial with a crossover study design aimed to compare the effect of stable supportive to flat flexible walking shoes on medial tibiofemoral joint contact force (MTCF) in individuals with medial knee osteoarthritis and varus malalignment. Waveforms were analyzed using parametric mapping with repeated measures ANOVA. Discrete outcomes were compared using repeated measures multivariate ANOVA models. Statistical parametric mapping showed a statistically significant difference between the two conditions. Loading impulse was significantly lower in supportive shoes compared to flexible ones.

Minor revisions:

1- Abstract: Improve the clarity of the objective statement. "To compare the effect of stable supportive shoes to flat flexible walking shoes...".

2- Table 1: The notation for mean +/- SD has not been utilized to display the values in the table.

3- The standard statistical term for average is mean.

Reviewer #2: COMMENT 1:

Thank you for the opportunity to review this manuscript. The manuscript " The immediate effect of flat flexible vs stable supportive walking shoes on tibiofemoral contact force in people with varus-malaligned medial knee osteoarthritis (PONE-D-21-29346)" contains interesting data that allow to understand the possible advantages and disadvantages of using different types of shoes in people with varus-malaligned medial knee osteoarthritis, from a complex biomechanical point of view and with with an important innovation using a novel insight from our use of neuromusculoskeletal modelling is the evaluation of the absolute muscular and external load components of MTCF.

I think these data would make a nice contribution to our knowledge on the type of shoes recommendation, based on biomechanics responses and would fit into PlosOne. The experimental approach and the methods of the study seem mostly sound, especially the idea about the integrative view of different biomechanical signals: kinematic, kinetic and electromyographic.

However, in the current state I consider the manuscript partly incomprehensive.I have several rather general suggestions because I think the manuscript requires several important changes to improve the overall scientific question.

TITLE AND RUNNING TITLE

COMMENT 2:

Title: I would like to suggest a change to the title, in order to make it more understandable: “Tiobiofemoral contact force differences between flat flexible and stable supportive walking shoes in people with varus-malaligned medial knee osteoarthritis.”

COMMENT 3:

Running title suggestion: Comparing tibiofemoral loads with different types of shoes in varus-malaligned osteoarthritis.

ABSTRACT

COMMENT 4:

Review: “Lower body motion”. In methods, full-body appears.

COMMENT 5:

Add the information that the tests were performed at a self-selected walking speed.

COMMENT 6:

Suggestion to change the text of the results: from “Statistical parametric mapping showed a significant effect between the two test conditions (p=0.001), with post-hoc tests showing lower MTCF in stable supportive compared to flat flexible shoes during 5-18% of stance phase.”; to “Statistical parametric mapping showed lower MTCF in stable supportive compared to flat flexible shoes during 5-18% of stance phase.”

INTRODUCTION

COMMENT 7:

Insert information regarding incidence rates of medial knee osteoarthritis to further justify the clinical relevance of the specific choice of this research sample.

COMMENT 8:

Insert a sentence at the end of the first paragraph that serves as a link to develop the subject about the type of shoe.

COMMENT 9:

Start a new paragraph after the end of line 49.

COMMENT 10:

Suggestion for changing the goal text: from “The aim of this study was to compare the effects of flat flexible and stable supportive walking shoes on MTCF waveforms using and discrete measures of MTCF and MTCF loading rate in people with medial knee OA and varus malalignment.”; to “The aim of this study was to compare the effects of flat flexible and stable supportive walking shoes on MTCF waveforms, discrete measures of MTCF and MTCF loading rate in people with medial knee OA and varus malalignment.”

COMMENT 11:

Rewrite the sentence (line 36 – 41), it is too long. Split into two sentences or increase the brevity.

COMMENT 12:

Does the word “surrogate” (line 52) have any other synonyms? What does it really mean? As it is written, it seems to me an alternative measure to some “gold standard” measure. If so, what would this “gold standard” measure be?

Here I take this opportunity to apologize for any failure in English, as it is not my native language.

METHODS

COMMENT 13:

I suggest rewriting the subtopics in order to separate and specify the “step-by-step” of data acquisition and the “step-by-step” of data processing. The text is a little confusing about this.

COMMENT 14:

Check spelling of the word “familiarisation” (line 134)

COMMENT 15:

What is the analysis area size, especially the distance traveled between two photoelectric beams? Was there any space for initial acceleration and final deceleration outside the collection area between the two photoelectric beams?

COMMENT 16:

There is important confusion in the description of the kinematic method used. It is necessary to explain in detail which reference was used for the construction of the kinematic model, since sixty-seven reflective markers were used, and the presented reference (number 27) uses a simpler model, only with lower-limb markers: “.. .using each thigh, shank, and foot segment cluster...".

COMMENT 17:

It is necessary to describe the processing of kinematic and kinetic data (obtained from the platforms). Was there any use of a standard Pipeline or was there a specific one created based on other references? What was the data filtering strategy? Was there a residual analysis of the signal? What software was used for these processes?

COMMENT 18:

Describe the diameter of the reflective markers.

COMMENT 19:

How was the Vicon system calibrated? What is the accepted Image error threshold?

COMMENT 20:

I particularly like the EMG data processing logic due to the normalization of the maximum EMG values.

COMMENT 21:

Please, insert the reference to “standardized anatomical landmark coordinates” line 192.

RESULTS

COMMENT 22:

I like the presentation of the results, however I believe that the information contained in the supplementary tables could be added to the main text, even considering that there is repeated information

COMMENT 22:

The figures that appear in the text file are in low resolution, however the images attached to the links are in good resolution.

COMMENT 23:

Please clarify how the data acquisition was carried out in individuals with unilateral OA, considering that 54% of the sample had unilateral symptoms?

DISCUSSION

COMMENT 24:

I suggest a general rewrite in the discussion, because in general it brings a lot of information about your results and speculations than a real discussion with other articles, looking for the mechanisms responsible for the answers found, whether they are in agreement or disagreement, and their reasons.

COMMENT 25:

I suggest deleting the first two sentences, adding a review of the main objective of the study and the main findings at the beginning of the first paragraph.

COMMENT 26:

Lines 350-352 “Inspection of Figure 3 waveforms suggest that at peak MTCF the muscle component to MTCF was higher, and the external component to MTCF lower, in flat flexible shoes compared to stable supportive shoes.”. It is an important result that goes in the opposite direction to what is presented at the conclusion of the study, as I understand that this relationship brings an important advantage to the flat flexible shoes. A more consistent discussion of this relationship is needed.

COMMENT 26:

A fundamental discussion point can be developed from a practical and evolutionary view. It is important to consider that, normally, since we started walking, we use shoes that can be considered as stable supportive shoes and the use of flat flexible shoes may not provide the most natural condition for the development of walking. I strongly suggest reading the article. Lieberman DE, Venkadesan M, Werbel WA, Daoud AI, D'Andrea S, Davis IS, Mang'eni RO, Pitsiladis Y. Foot strike patterns and collision forces in habitually barefoot versus shod runners. Nature. 2010 Jan 28;463(7280):531-5. doi: 10.1038/nature08723. PMID: 20111000.

A link to this discussion can be made from the idea presented in lines 388-389 "Study of longitudinal responses of MTCF to footwear needs to be developed to understand any muscle or gait adaptations.", considering the "biomechanical mechanisms" under a perspective evolution of the type of shoes used

REFERENCES

COMMENT 27:

Review the list of references, especially regarding the writing pattern for citing each journal, for example: “Gait Posture”; “Gait & Posture”. As well as the abbreviations of each journal.

6. PLOS authors have the option to publish the peer review history of their article (what does this mean?). If published, this will include your full peer review and any attached files.

Reviewer #1: No

Reviewer #2: No

---

## [Author Response · Author response to Decision Letter 0]

25 Feb 2022

Please refer to the "response to reviewers" document

---

## [Decision Letter · Decision Letter 1]

19 May 2022

Tibiofemoral contact force differences between flat flexible and stable supportive walking shoes in people with varus-malaligned medial knee osteoarthritis: a randomized cross-over study

PONE-D-21-29346R1

Dear Dr. Hall,

We’re pleased to inform you that your manuscript has been judged scientifically suitable for publication and will be formally accepted for publication once it meets all outstanding technical requirements.

Kind regards,

Leonardo A. Peyré-Tartaruga, Ph.D.

Academic Editor

PLOS ONE

Reviewer #1: All comments have been addressed

Reviewer #2: All comments have been addressed

2. Is the manuscript technically sound, and do the data support the conclusions?

Reviewer #1: (No Response)

Reviewer #2: Yes

3. Has the statistical analysis been performed appropriately and rigorously? 

Reviewer #1: (No Response)

Reviewer #2: Yes

4. Have the authors made all data underlying the findings in their manuscript fully available?

Reviewer #1: (No Response)

Reviewer #2: Yes

5. Is the manuscript presented in an intelligible fashion and written in standard English?

Reviewer #1: (No Response)

Reviewer #2: Yes

6. Review Comments to the Author

Reviewer #1: (No Response)

Reviewer #2: I thank the authors for the answers to all questions and suggestions. In fact, they carried out detailed work that significantly improved the quality of the paper, especially the corrections made in the methods and discussion chapters.

7. PLOS authors have the option to publish the peer review history of their article (what does this mean?). If published, this will include your full peer review and any attached files.

Reviewer #1: No

Reviewer #2: **Yes: **Henrique Bianchi Oliveira

---

## [Editor Report · Acceptance letter]

23 May 2022

PONE-D-21-29346R1 

Tibiofemoral contact force differences between flat flexible and stable supportive walking shoes in people with varus-malaligned medial knee osteoarthritis: a randomized cross-over study 

Dear Dr. Hall:

I'm pleased to inform you that your manuscript has been deemed suitable for publication in PLOS ONE. Congratulations! Your manuscript is now with our production department. 

Kind regards, 

on behalf of

Professor Leonardo A. Peyré-Tartaruga 

Academic Editor

PLOS ONE